# Current Challenges of Methylation-Based Liquid Biopsies in Cancer Diagnostics

**DOI:** 10.3390/cancers16112001

**Published:** 2024-05-24

**Authors:** Tomas Rendek, Ondrej Pos, Terezia Duranova, Rami Saade, Jaroslav Budis, Vanda Repiska, Tomas Szemes

**Affiliations:** 1Institute of Medical Biology, Genetics and Clinical Genetics, Faculty of Medicine, Comenius University, 811 08 Bratislava, Slovakia; vanda.repiska@fmed.uniba.sk; 2Geneton Ltd., 841 04 Bratislava, Slovakia; ondrej.pos@uniba.sk (O.P.); jaroslav.budis@geneton.sk (J.B.); tomas.szemes@geneton.sk (T.S.); 3Comenius University Science Park, 841 04 Bratislava, Slovakia; duranova33@uniba.sk; 42nd Department of Gynaecology and Obstetrics, Faculty of Medicine, Comenius University, 811 08 Bratislava, Slovakia; rami.saade@fmed.uniba.sk

**Keywords:** cfDNA, methylation, liquid biopsy, cancer diagnostics

## Abstract

**Simple Summary:**

This review discusses the importance of advancements in cancer diagnosis and treatment, emphasizing the need for early detection methods. It highlights the role of DNA methylation, an epigenetic change, as a promising marker for detecting cancer. This review explains how DNA methylation patterns differ between cancerous and healthy cells, influencing gene expression and contributing to genomic instability. It also delves into the challenges and methods associated with detecting DNA methylation, particularly in cell-free DNA (cfDNA) from bodily fluids, known as liquid biopsies. Different tests for single-cancer detection, such as colorectal, lung, and bladder cancer, as well as multi-cancer detection tests, are discussed along with their methodologies and clinical applications. Biological factors influencing DNA methylation, such as age, gender, and environmental influences, are also explored. The conclusion emphasizes the potential of cfDNA methylation analysis in cancer diagnostics but acknowledges the current limitations and challenges, highlighting the need for further research and advancements in methodology and understanding.

**Abstract:**

In current clinical practice, effective cancer testing and screening paradigms are limited to specific types of cancer, exhibiting varying efficiency, acceptance, and adherence. Cell-free DNA (cfDNA) methylation profiling holds promise in providing information about the presence of malignity regardless of its type and location while leveraging blood-based liquid biopsies as a method to obtain analytical samples. However, technical difficulties, costs and challenges resulting from biological variations, tumor heterogeneity, and exogenous factors persist. This method exploits the mechanisms behind cfDNA release but faces issues like fragmentation, low concentrations, and high background noise. This review explores cfDNA methylation’s origins, means of detection, and profiling for cancer diagnostics. The critical evaluation of currently available multi-cancer early detection methods (MCEDs) as well as tests targeting single genes, emphasizing their potential and limits to refine strategies for early cancer detection, are explained. The current methodology limitations, workflows, comparisons of clinically approved liquid biopsy-based methylation tests for cancer, their utilization in companion diagnostics as well as the biological limitations of the epigenetics approach are discussed, aiming to help healthcare providers as well as researchers to orient themselves in this increasingly complex and evolving field of diagnostics.

## 1. Introduction

Cancer is poised to become the primary cause of death worldwide, highlighting the urgency for advancements not only in more impactful treatments [1] but also in new effective diagnostic methods with high clinical utility for early cancer detection [2,3]. The detection of cancer before stage IV could potentially reduce cancer-related deaths by at least 15% within a five-year period [1,4]. The emerging paradigm shift in disease diagnosis involves a growing reliance on molecular characterization, moving beyond conventional clinical and symptom-based assessments. While genetic alterations and transcription signatures were initially proposed as biomarkers, their clinical use is constrained by issues of reproducibility and accuracy. Consequently, attention has shifted towards exploring epigenetic changes as a viable alternative for disease diagnosis [5].

DNA methylation, a prominent epigenetic change in humans, has been heavily studied in cancer [6,7]. It plays crucial roles in gene expression regulation, X-chromosome inactivation, genomic imprinting, and allele-specific expression [8,9,10]. DNA methylation primarily targets cytosine residues in CpG dinucleotides, typically clustered in CpG islands. These islands are primarily located at the 5′ end of genes and represent approximately 60% of the promoter regions of human genes. Approximately 30,000 CpG islands are present in the human genome, and under physiological conditions, 70–80% of them are methylated [11]. The dysregulation of DNA methylation is linked to various pathological conditions including cancer, developmental disorders, and aging. The methylomes of cancer cells differ from healthy cells and may be used to distinguish between different cancer types, as some genes develop tissue-specific DNA methylation patterns. Since these changes can be assessed in various body fluid samples, DNA-methylation-based liquid biopsies are widely recognized as promising less-invasive approaches for prognostic assessment as well as for the identification of premalignant/early cancer [12]. One of the significant advantages of a liquid biopsy is its ability to follow the progression of malignant tumors, both primary and metastatic, and to identify when their recurrence has occurred [13]. In addition, it is cheaper and less invasive, thus increasing patient acceptance [14]. However, DNA methylation detection is associated with certain limitations that must be carefully considered. Cell-type composition, cellular contamination, technical problems, allele- and strand-specific DNA methylation, and random loss of DNA methylation at a specific locus are examples of potential sources of heterogeneity [15,16].

## 2. DNA Methylation and Cancer

Cancer development can be described as an interplay between genetic and epigenetic alterations, with changes in DNA methylation patterns being among the earliest, most consistent, and prevalent indicators [17]. Numerous theories have been advanced about the reason for aberrant methylation in cancer cells, primarily revolving around two fundamental concepts: differential fitness and targeted selection [18]. The differential fitness theory proposes that abnormal DNA hypermethylation initiates with the indiscriminate dissemination of DNA methylation across the genome, potentially stemming from dysregulation of the DNA methylation apparatus. Such methylation occurrences predominantly manifest within gene promoters responsible for constraining cell viability and growth. The alternative theory suggests that hypermethylation arises from the anomalous localization of DNMTs to specific genomic regions and/or that these regions inherently harbor cis-regulatory elements, rendering them more conducive to de novo DNA methylation [19]. The methylome of malignant cells in comparison with normal cells is characterized by two types of general changes: global hypomethylation or focal hypermethylation at CpG islands. Since DNA methylation affects chromatin structure and function, these changes lead to an abnormal regulation of gene expression and the derepression of imprinted genes and retrotransposons and contribute to the occurrence of somatic mutations, resulting in overall genomic instability [11,12,20,21]. Genes with hypomethylated promoter regions have higher expression levels, whereas the acquisition of methylation in the initially non-methylated promoter region can inhibit the ability of transcription factors to bind, resulting in transcription down-regulation or gene silencing [22]. Different genes are methylated at various stages of cancer development, influencing oncogene activation and tumor-suppressor gene inactivation [11,12,20]. It is well known that focused promoter hypermethylation causes the transcriptional suppression of tumor-suppressor genes in cancer [14]. On the contrary, global methylation levels in benign tissue were found to be considerably higher than in malignant tissue [23], indicating that hypermethylated CpG sites are more prone to mutations than their regularly methylated counterparts [24]. Hypomethylation of repeated DNA sequences or proto-oncogenes has a markedly negative impact on chromosomal stability and cellular function [25]. Moreover, it has been proposed that aneuploidy results from the excessive hypomethylation of centromeric and pericentromeric satellite sequences, which is frequently observed in a variety of malignancies [12]. Meta-analyses of global DNA hypomethylation in a variety of malignancies have revealed a correlation between the degree of hypomethylation and cancer stage, confirming its role in carcinogenesis. On an overall methylation level in malignant cells, colorectal cancer (CRC) is very well studied, where a comparison between CRC cells and normal colorectal epithelial cells revealed that over 10% of protein-coding genes exhibit differential methylation [26]. Several methylation markers have been found to date that are related to a wide spectrum of cancers, including breast cancer (e.g., hypermethylated *APC*, *FOXA1*, and *RASSF1A*) [27], esophageal cancer (e.g., hypermethylated *TP53*, *CDKN2A*, *CTNNB1*, and *APC*) [28,29], gastric cancer (e.g., hypermethylated *p16*, *RUNX3*, *RASSF10*, *APC*, and hypomethylated *LINE1*) [28,29], lung cancer (e.g., hypermethylated *DAL-1*, *EPHB6*, *HS3ST2*, *MGMT*, and hypomethylated *ELMO3*) [30], CRC (e.g., hypermethylated *SEPT9*, *MGMT*, *SDC2*, *N-Myc*, and *APC*) [30,31], and hepatocellular carcinoma (e.g., hypermethylated *HOXA1*, *EMX1*, *ECE1*, and *PFKP*) [32].

## 3. Methylation Detection in Cell-Free DNA

Cell-free DNA (cfDNA) fragments released from cells preserve genetics and epigenetics background information from the original tissue [33]. Apart from primary structure and epigenetic marks, the increased concentration of cfDNA in the body fluids of cancer patients compared to healthy individuals makes it a promising marker of neoplasms [34,35]. Results have clearly shown that it is tumor-type- and progression-specific [36], allowing for the real-time monitoring of disease dynamics and treatment response. Among the most common epigenetic modifications of cfDNA is methylation, which has received increased attention in recent years. The results of experiments indicate that cfDNA isolated from two different tumors is more likely to vary in somatic genetic mutations than in epigenetic information, which remains consistent in many cases, again suggesting the potential of cfDNA methylation [37].

Until now, the detection of epigenetic alterations in cancer cells has been carried out mainly from solid tumors, but due to the invasiveness and complexity of samplings, over time, research is starting to move to liquid biopsy sampling as a reliable alternative. [38,39].

Liquid biopsy represents a method focused on reducing the invasiveness of the biological sampling from the patient. For liquid biopsy body fluids, blood and its individual components, cerebrospinal fluid, urine, or saliva are commonly used to analyze cfDNA or circulating tumor DNA (ctDNA) as well as other cancer-related analytes, e.g., exosomes, miRNAs, and proteins [40].

The principle of ctDNA detection in bodily fluids is based precisely on the methylation profile of the DNA fragment, which is typical for tumor cells [41,42]. Hypermethylation occurs in ctDNA derived from a wide range of tumor types, such as lung [43], breast [43,44], colon [45], or prostate [46] neoplasms. Since DNA fragments from necrotizing and apoptotic tumor cells are directly excreted into the body during the tumor process, methylated ctDNA is considered tumor-specific because of its primary source, which provides additional valuable information in diagnostic applications [47].

### 3.1. Methods for Analyzing DNA Methylation Status

There are two main approaches for detecting methylated sites of ctDNA regarding the range of the target DNA sequence: (I) whole-genome analysis and (II) methods focused on specific DNA loci. Generally, the manual and technical difficulty is higher for methods analyzing the whole genome, and these methods suddenly require validation at the level of the selected loci [47,48].

There are three categories of methods in terms of the molecular mechanism for DNA-methylation detection: (I) methods based on the detection of sodium bisulfite conversion, (II) methods using the interaction of antibodies against 5mC with methylated sections of DNA, and (III) methods using enzymatic cleavage. Among the above-stated methods, antibody–DNA interaction methods are widely used, e.g., enzyme-linked immunosorbent assay (ELISA) assays, mainly due to their availability and easy handling. There are several currently commercially available kits, preponderantly based on the sandwich ELISA principle, which is rough and, unfortunately, does not provide specific information about methylation. In addition, their further disadvantage is high variability together with cross-reactivity [49], which ultimately indicates the improbable use of ELISA-based assays in the detection of DNA methylation in clinical practice.

Methods based on the selective cleavage of DNA using methylation-sensitive restriction enzymes (MSREs) capable of cleaving unmethylated sections of CpG islands while leaving methylated fragments uncleaved are also often used [50]. With the subsequent use of qPCR, it is a rapid, reproducible, and robust method that can also be used in combination with sequencing, either Sanger, bisulfite, or even next-generation sequencing (NGS), while the last-mentioned method is accompanied by increased complexity and costs.

Last but not least, the method of DNA methylation detection using sodium bisulfite conversion is currently considered a gold standard in DNA methylation screening. The treatment of the DNA sample with sodium bisulfite converts unmethylated cytosine to uracil and then converts uracil to thymine during subsequent methylation-specific PCR (MSP) [51]. Primers used for MSP are designed as methylation-specific, complementary to unconverted 5-methylcytosines, or, conversely, to the converted sequence. It is also applicable with high sensitivity for sequences with a low density of methylation, while a high density of CpG islands is directly related to increasing the specificity of this test [52]. In the case of quantitative methylation-specific PCR (qMSP), in addition to specifically designed primers, methylation-specific fluorescent reporter probes are also used, which allow not only for very sensitive detection but also for the quantification of the DNA methylation level [53].

In clinical practice, genome-wide bisulfite sequencing is practically inapplicable, mainly due to its difficulty and high cost; thus, currently, it serves primarily for research purposes to map epigenome methylation [54]. The analytical boundaries of genomic ctDNA assays are commonly assessed through the measurement of the variant allele frequency that denotes the proportion of sequencing reads containing tumor-specific mutations relative to the total number of sequencing reads aligning with the corresponding genomic loci [55].

A targeted bisulfite sequencing of precise loci is deemed more effective [56], offering a reliable tool for validating data obtained, for example, by array-based methods. Array-based methods enable the screening of methylation on a genome-wide scale, but similar to whole-genome sequencing, this method also has a relatively high error rate, and targeted bisulfite sequencing could serve as one of its validation options in clinical practice [56,57].

Although improvements in amplicon-based NGS have enhanced reliability and sensitivity, its utility is constrained to established hotspots. Additionally, panels are comparatively more affordable than capture NGS [58]. Methylation signatures of ctDNA identified through various bioinformatics models may exhibit inconsistency, resulting in disparities or partiality within their predictive models [59]. Furthermore, each assay developed by different research teams has its criteria for defining positive or negative results, as well as distinct methodologies for ctDNA extraction, analysis, and quantification [60]. An overview of the current laboratory methods for methylation detection and their respective characteristics, including their major advantages and disadvantages, can be found in Appendix A.

### 3.2. Clinically Approved Methylation-Based Cancer Screening

In clinical practice, the most frequently used methodology is the bisulfite conversion method, followed by PCR or sequencing; thus, the majority of clinically approved methylation-based screenings are based on this principle [61]. Below, we summarize the currently available tests implementing a methylation and liquid biopsy approach, distinguishing between cancer-specific and multi-cancer detection. A list of the currently available tests, along with their targets, specificity, sensitivity, and source of cfDNA, can be found in Table 1, and their visual overview is shown in Figure 1.

#### 3.2.1. Single-Cancer Detection Tests

##### Colorectal Cancer

The majority of commercially available methylation-based tests designed for CRC screening use stool samples as a source of DNA, such as Colovantage^®^ [62] CologuardTM [62,63], and ColoSureTM [64], but only a few blood-based tests are currently available on the market.

The Epi proColon^®^ 2.0 CE (Epigenomics AG, Berlin, Germany) test represents a second-generation screening tool designed to detect the methylation status of the gene *Septin9* from plasma-derived cfDNA based on PCR. The test demonstrates an overall sensitivity in detecting CRC of 68.2% across all stages, alongside a specificity of 79.1% [65]. This test is accessible on the European market and in various other regions, including China [66]. Compared with the original 1.0 version of the test, the newest tool boasts enhanced sensitivity and specificity compared to its predecessor, which analyzed samples in duplicate rather than triplicate [67].

Another qPCR-based diagnostic test is COLVERA, designed to identify DNA methylation status in the *BCAT1* and *IKZF1* genes that exhibit hypermethylation in 95% of CRC tissue. COLVERA has demonstrated enhanced sensitivity for detecting recurrent disease compared to using carcinoembryonic antigen across various clinical populations [68]. The clinical performance of this test varies, as reported in a validation study by Murray et al., with sensitivity and specificity of 73.1% and 89.3%, respectively [69]. Another study was conducted on a cohort of 322 individuals, where the COLVERA test reached an overall sensitivity of 59.3% (95% CI: 38.8–77.6) and specificity of 98.3% (96.1–99.5) [68].

A different approach applies the Nu.Q™ Colorectal Cancer Test, aiming to reduce the number of unnecessary colonoscopies. The test was developed on the principle of global hypomethylation present in cfDNA of a CRC origin [70] and utilizes the ELISA method to analyze the methylation status of nucleosomes present in the serum, with sensitivity at 75% and with a 70% specificity [71,72].

##### Lung Cancer

EarlyTect^®^ L is a validated epigenetic biomarker test centered on *PCDHGA12* methylation status in serum, demonstrating reliable clinical sensitivity and specificity in detecting lung cancer. A study involving 522 participants has confirmed its capability to identify lung cancer, exhibiting a sensitivity of 77.8% and specificity of 92.3%. The test employs standard qPCR instrument systems to analyze epigenetic markers present in 2 mL serum samples [73,74].

Another test able to distinguish malignant lung disease is Epi ProLung. A qPCR method evaluates the DNA methylation levels of *SHOX2*, *PTGER4*, and the reference gene *ACTB* using plasma samples obtained from individuals both with and without malignant disease. The validation process yielded a sensitivity of 78% and a specificity of 96% [75].

##### Bladder Cancer

Urodiag^®^ employs Mutated-Allele-Specific Oligonucleotide PCR (MASO-PCR) for the simultaneous amplification of *SLIT2*, *SEPTIN9*, and *HS3ST2* methylated genes [76] for the surveillance of non-muscle-invasive bladder cancer. In a surveillance setting, the test exhibited a sensitivity of 94.5% and a specificity of 75.9% [77].

AssureMDx (MDxHealth) is another qPCR-based assay that integrates the genetic mutation status assessment of *FGFR3*, *TERT*, and *HRAS* with the examination of epigenetic patterns in *OTX1*, *ONECUT2*, and *TWIST1* within exfoliated urinary cells. By employing a logistic regression analysis of these markers along with patient age, a diagnostic model demonstrated a sensitivity of 97% and specificity of 83% according to the initial findings [78].

The Bladder Epicheck (Nucleix) assay is a commercially available qPCR-based method designed to evaluate DNA methylation patterns in urinary exfoliated cells across 15 genomic regions. In the surveillance context, the test exhibited a sensitivity of 90% and a specificity of 83% [79], while in a larger-scale prospective study, it reached a sensitivity of only 68.2% with a specificity of 88% [79,80].

The UroMark assay presents another promising method for detecting bladder cancer through residual DNA in voided urine. This assay focuses on 150 CpG loci utilizing NGS bisulfite sequencing. It successfully differentiated primary bladder cancer from non-bladder cancer urine samples with a sensitivity and specificity of 98% and 97% [81].

EarlyTect^®^ B is a qPCR-based method for analyzing multiple genomic and epigenomic targets in cfDNA extracted from urine samples [70,82] and is commercially available in East Asia.

Other available tests for bladder cancer screening utilizing exfoliating cells in urine are Urifind (Anchor Dx), providing a 91.7% sensitivity and 77.3% specificity [83], and Bladmetrix (SDU), with a 92.1% sensitivity and 93.3% specificity [84].

##### Liver Cancer

As imagining and biochemical markers remain substandard methods for the screening of hepatocellular carcinoma (HCC), an epigenetic-based blood test, HCCBloodTest (Epigenomics AG), has been developed, utilizing NGS bisulfite sequencing targeting the methylation status of the *SEPT7* gene [85], achieving, in clinical study NCT03804593, a sensitivity and specificity of 76.7% and 64.1%, respectively.

Another test focusing specifically on hepatic malignancies is IvyGene Liver, with a sensitivity of 80% and a specificity of 86% [86].

It is worth mentioning that methylation-based noninvasive tests have also been developed for the detection of cervical cancer based on the methylation status of the ZNF582 gene, namely, Cervi-M^®^ [87], as well as for head and neck cancer, namely, Oral-M^®^, with additional *PAX1* gene methylation detection. [88,89]. Since both tests are swab-based, they do not qualify as liquid biopsy methods, and they are not further studied in this review, although the Cervi-M^®^ test is discussed to possibly utilize tampon blood as a source of DNA, thus qualifying as a liquid biopsy [88,89].

#### 3.2.2. Multi-Cancer Detection Tests

Lately, there has been a surge in research addressing the screening gap through the exploration of multi-cancer early detection (MCED) tests designed to generate cancer signal without offering a conclusive diagnosis [90]. Leveraging blood-based ctDNA for the simultaneous detection and localization of multiple cancer types may offer a solution to this significant unmet demand. Implementing such a multi-cancer detection strategy in large-scale population screening necessitates high specificity, clinically relevant sensitivity, and precise tissue of origin identification to manage the scale, expenses, and intricacies involved in evaluating asymptomatic patients [91,92].

##### Galleri^®^ Test

The Galleri^®^ test (GRAIL) is the first commercially available MCED able to detect more than 50 types of cancer [93]. The test utilizes a focused NGS bisulfite sequencing methylation assay, covering over 100,000 distinct regions and more than 1,000,000 cytosine–guanine dinucleotides coupled with machine learning techniques. The system can identify cancer signals across various cancer types and accurately predict the origin of these signals, known as Cancer Signal Origin (CSO) [94,95].

Its ability was investigated in the well-known PATHFINDER study (NCT04241796), a prospective clinical trial across seven US health networks that enrolled 6662 adults aged 50 or older without cancer symptoms for MCED testing from 12 December 2019 to 4 December 2020. Among the 6621 participants with analyzable results, 4204 (63.5%) were women and 2417 (36.5%) were men. A cancer signal was detected in 92 (1.4%) participants, with 35 (38%) receiving a cancer diagnosis (true positives) and 57 (62%) not diagnosed with cancer (false positives). Excluding two participants whose diagnostic assessments began before the MCED test results, the median time to diagnostic resolution was 79 days (IQR 37–219), with 57 days (33–143) in true-positive participants and 162 days (44–248) in false-positive participants. Most participants underwent both laboratory tests (26 [79%] of 33 true-positive participants and 50 [88%] of 57 false-positive participants) and imaging (30 [91%] of 33 true-positive participants and 53 [93%] of 57 false-positive participants) [96].

Another randomized controlled trial, called NHS-Galleri (ISRCTN91431511), is being conducted to evaluate the effectiveness of a blood test in detecting cancers early, potentially reducing the occurrence of late-stage cancers. Over 140,000 individuals from the general population of England in the age span of from 50 to 77 years who have neither been diagnosed with nor are under investigation for cancer have been enrolled in the NHS-Galleri trial. Blood samples are being collected up to three times: initially upon study enrollment and subsequently at 12- and 24-month intervals [97]. The results of the study are expected to be released in the spring or summer of 2024 [98]. To this day, only limited data from ongoing clinical trials are available. The revealed data project the positive predictive value (PPV) of this test to be below 10%, a level that may not adequately facilitate an effective screening initiative, along with modest sensitivities for early-stage cancer (17% for stage I, 40% for stage II, and 27.5% for stages I-II) [95]. A good overview of sensitivities and PPV in a setting of 99% specificity provides a reply to published data by Pons-Belda et al. [99].

##### PanSeer

The PanSeer test employs a comparable methodology to the Galleri test, encompassing 11,787 CpG sites spanning 595 genomic regions [94]. In the Taizhou Longitudinal Study (TZL), plasma samples from 123,115 initially healthy individuals were collected and stored for future analysis, with subsequent cancer monitoring. The preliminary findings from PanSeer, a noninvasive blood test utilizing ctDNA methylation, were evaluated using TZL plasma samples from 605 asymptomatic individuals, including 191 later diagnosed with various cancers within four years. PanSeer exhibited a detection rate of 88% (95% CI: 80–93%) in post-diagnosis patients with a specificity of 96% (95% CI: 93–98%) and 95% (95% CI: 89–98%) in asymptomatic individuals later diagnosed, warranting further longitudinal studies for confirmation [71,100].

##### IvyGene

IvyGene, developed by the Laboratory for Advanced Medicine, is a diagnostic test designed to assess the levels of four common cancers (breast, colon, liver, and lung) by analyzing the methylation status of cfDNA extracted from patients’ blood samples using a panel of 46 markers [101]. Positioned as a supplementary clinical tool, physicians can order this test to complement patient evaluations, and it is currently available in the USA as a direct-to-consumer service [102]. Currently, the test is offered as IvyGeneCORE, targeting four mentioned cancers, producing a sensitivity of 84% and specificity of 90% [86].

##### CancerRadar

A recently developed MCED test (not commercially available yet) is CancerRadar, which combines cfMethyl-Seq for genome-wide cfDNA methylation profiling, offering an over 12-fold enrichment compared to whole-genome bisulfite sequencing in CpG islands with a computational platform for extracting diverse methylation data and patient diagnosis [94,103]. In a study involving 408 colon, liver, lung, and stomach cancer patients and controls, a specificity of 97.9% and sensitivities of 85.9% for detecting all-stage cancer and 81.4% for early-stage (I and II) cancers have been achieved. Additionally, accuracies of 89.6% for identifying the tissue of origin of all-stage cancer and 84.4% for early-stage cancer were attained [94].

**Table 1 cancers-16-02001-t001:** Overview of available methylation-based liquid biopsy tests analyzing cell-free DNA for cancer detection with their respective focus, specificity, sensitivity, and source of cfDNA.

Test	Cancer Type	Specificity	Sensitivity	Sample Type	Ref.
Epi proColon^®^ 2.0 CE	Colorectal Cancer	79.1%	68.2%	Blood (plasma)	[65]
COLVERA	Colorectal Cancer	89.3%	73.1%	Blood (plasma)	[69]
Nu.Q™ Colorectal Cancer Test	Colorectal Cancer	70%	75%	Blood (serum)	[71,72]
EarlyTect^®^ L	Lung Cancer	92.3%	77.8%	Blood (serum)	[73,74]
Epi ProLung	Lung Cancer	96%	78%	Blood (plasma)	[75]
Urodiag^®^	Bladder Cancer	75.9%	94.5%	Urine	[77]
AssureMDx	Bladder Cancer	83%	97%	Urine	[78]
Bladder Epicheck	Bladder Cancer	83%	90%	Urine	[79]
UroMark	Bladder Cancer	97%	98%	Urine	[81]
EarlyTect^®^ B	Bladder Cancer	N/A	N/A	Urine	[70,82]
Urifind	Bladder Cancer	77.3%	91.7%	Urine	[83]
Bladmetrix	Bladder Cancer	93.3%	92.1%	Urine	[84]
HCCBloodTest	Liver Cancer	64.1%	76.7%	Blood	[85]
IvyGene Liver	Liver Cancer	86%	80%	Blood	[86]
Galleri^®^ Test	Multi-cancer Early Detection	N/A *	N/A *	Blood	[96]
PanSeer	Multi-cancer Early Detection	96%	88%	Blood	[71,100]
IvyGeneCORE	Breast, Colon, Liver, and Lung Cancer	90%	84%	Blood	[86]
CancerRadar	Multi-cancer Early Detection	97.9%	85.9%	Blood	[94]

* Specificity and sensitivity vary among cancer types and stages, as discussed in Section 3.2.

###### 3.2.3. Methylation-Based Methods in Companion Diagnostics

The variability in pharmacotherapy can be substantial, often attributed to the heterogeneity of cancer, posing the necessity to develop predictive assays based on specific biomarkers to guide targeted therapy use, known as companion diagnostics [104]. Epigenomics holds significant potential in predicting treatment response and cancer susceptibility. An exemplary illustration of the practical application of DNA methylation in personalized medicine is demonstrated through the utilization of *MGMT* promoter hypermethylation. This method effectively categorizes the glioblastoma patients in clinical trials and aids in determining their response to alkylating agents. MGMT, a pivotal DNA repair enzyme, plays a crucial role in eliminating specific and abnormal DNA adducts [105]. As this method is not liquid-biopsy-based and invasive biopsy is necessary, its broad clinical application remains challenging [106].

For breast cancer methylation-based companion diagnostics, interesting research was conducted by Mastoraki et al., assessing the *ESR1* gene methylation status as a predictive biomarker, as gene methylation was correlated with non-responsiveness to the everolimus/exemestane treatment. Further investigation into *ESR1* methylation is warranted to assess its potential as a liquid-biopsy-based biomarker [107].

In brain tumors, the methylation status of *MGMT-STP27* was shown as a promising predictive biomarker to assess the response to a PCV chemotherapy regimen (procarbazine, CCNU, and vincristine) in anaplastic oligodendrogliomas and oligoastrocytomas [106,108,109]. For the purpose of decision-making, PyroMark Therascreen *MGMT* and the PyroMark Q96 CpG *MGMT* kits were developed [110]. Great promise for liquid-biopsy-based diagnostics in brain tumors lies in the utilization of cerebrospinal fluid, as discussed in a review by Eibl and Schneemann [111].

## 4. Biological Factors Influencing DNA Methylation

Apart from the technology-related limitations mentioned in Section 3.1, there are also biological and physiological factors influencing the methylation of somatic cells, which must be counted on, especially when assessing global methylation status for cancer diagnostics. Inter- and intra-tumor heterogeneity exists due to variability in the morphology, genetics, epigenetics, and phenotype of cell populations. Comprehending the variations in phenotypes requires knowledge of variations in DNA methylation variability amongst cell groups [112,113,114]. Heterogeneity within profiled cell populations is often ignored in research because they only look at average DNA methylation levels of individual CpGs. Liquid biopsies are more useful in such a situation, as tissue biopsies can capture only part of this heterogeneity [115].

The potential for false-positive or false-negative results due to methylation changes arises with aging, inflammation, or other non-cancerous conditions. Other physiological characteristics that may arise in particular clinical circumstances, such as a cardiac infarct, may also modify the epigenetic and biological aspects of cfDNA.

Individual differences in DNA methylation patterns can result from both hereditary and environmental influences. Variations in DNA segments or genes are associated with approximately 20% of the interindividual variation in DNA methylation patterns [15,16]. Dietary components that affect DNA methylation include folate absorption [116]. Age is another interesting factor of methylation, with studies referring to changing patterns in the methylation of tissues over time as a methylation clock successfully determining age [117]. Gender is also significantly correlated with tissue-specific DNA methylation levels [118,119], which is particularly important in autoimmune disorders with a strong sex bias. For example, X-chromosome DNA methylation variations determined in the manner of the original parent may account for the higher frequency of autoimmune disorders in women than in men [120].

Metabolic disorders, such as diabetes mellitus [121] and obesity [121,122], are associated with differential methylation phenotypes compared to healthy individuals.

## 5. Conclusions

While the utilization of cfDNA methylation analysis in cancer diagnostics holds significant promise, it is currently hindered by various biological limitations and technical challenges. Circulating DNA originating from rare tumor subpopulations might only be detectable in ctDNA at extremely low concentrations, posing challenges along with inter- and intra-tumor heterogeneity and the stage of its development, along with the factors of age, gender, environmental influences, and biological activities in normal cells introducing variability in methylation patterns, leading to misdiagnosis. In clinical practice, currently, there is a limited market of epigenetic-based liquid biopsy tests screening for the global presence of cancer in the body, and more clinical studies and improved performance for early-stage cancers must be achieved. This review discussed the most common techniques for methylation detection in cfDNA, their limitations, and their suitability for specific clinical use. For diagnostic tests focusing on methylation detection on a single or limited number of sites on the genome related to a particular cancer type, PCR remains the most robust, economical, and sensitive method. On the other hand, MCED tests, which are set to target multiple cancers or general cancer signals, have to utilize genome-wide methylation sequencing or array-based methods, coming with challenges associated with reproducibility and higher costs. From a cancer-type perspective, the biggest number of tests were developed for bladder cancer, despite being only the 10th most common cancer worldwide. For this cancer, no screening tests for the population at average risk have been developed to this day. Following these are the tests for colorectal cancer, where methylation-based liquid biopsy has to compete with already established and relatively inexpensive fecal occult blood tests (FOBTs). Following these are tests for lung cancer, a disease with the highest mortality and missing efficient and noninvasive screening tests. Cancer-specific liquid biopsy test limitations remain in terms of poor performance, a lack of clinical and post-market data, and education of clinicians in the possibility of their adoption, especially for patients refusing standard diagnostic methods. The use of MCED tests is usually limited to a certain number of cancers. Alternatively, they produce a tissue-specific or non-specific cancer signal, which can navigate healthcare providers for further testing. Their use for screening on the general public remains controversial, raising questions about the benefits, harms, and high costs involved, as seen from already ongoing trials. The standardization of pre-analytical techniques, improved assay methodologies, and an enhanced understanding of the biological mechanisms underlying methylation patterns are essential steps toward overcoming these challenges and realizing the full clinical potential of ctDNA methylation analysis in oncology, encouraging further basic as well as clinical research.

## Figures and Tables

**Figure 1 cancers-16-02001-f001:**
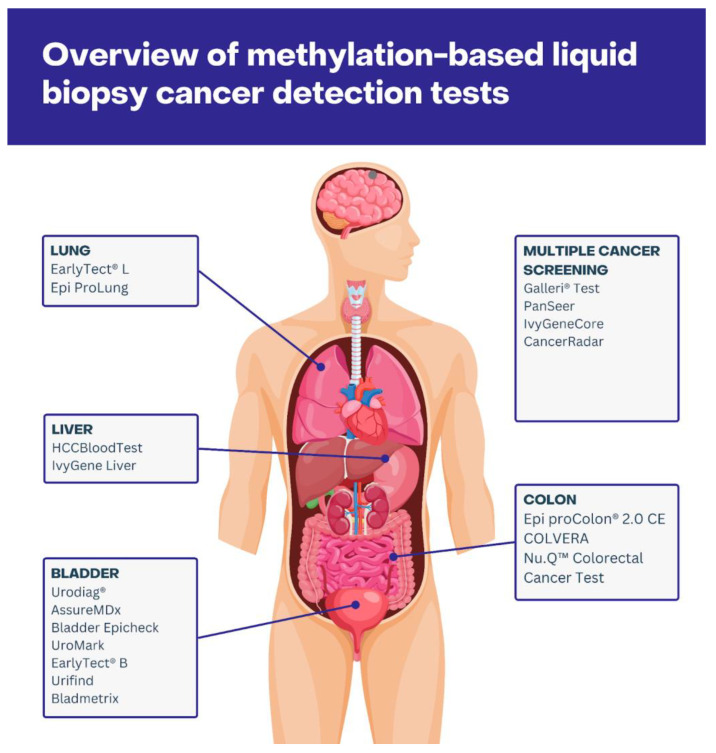
Visual overview of methylation-based liquid biopsy tests focusing on multi-cancer detection or detection of neoplasm in particular organs.

## Data Availability

The original contributions presented in the study are included in the Appendix A. Further inquiries can be directed to the corresponding author.

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
