# Peer review of "Current Challenges of Methylation-Based Liquid Biopsies in Cancer Diagnostics"

_cancers, 2024, doi:10.3390/cancers16112001_

Round 1

Reviewer 1 Report

Comments and Suggestions for Authors

This peer review is on a manuscript submitted for a special edition of “Current and Emerging Utility of Liquid Biopsy in Cancers: More than Surrogate Biomarkers (2nd Edition)“. Rendek and colleagues provide a comprehensive overview of methylation in cancer and how this knowledge on the epigenetic status of cancer-related genes can be applied to diagnose cancers earlier or more accurately using liquid biopsy, including better monitoring. The authors summarize the tests currently available for specific types of cancer and for screening multiple cancers. Limitations are discussed as well. The review is informative, easy to read, and addresses important developments for the intended readership. The manuscript may be further improved by addressing very minor issues:

  1. Line 17: Perhaps avoid the word “delves”, since this appears to be used often by ChatGPT.
  2. Line 44: Consider rephrasing "Cancer is poised to become the primary cause of death worldwide." This claim is debatable. Even the reference cited only refers to a specific age group in high-income countries. So, worldwide, cancer is far from becoming the primary cause of death.
  3. Line 46 "Detection of cancer before stage IV could potentially reduce cancer-related deaths by at least 15% within a five-year period" – This may be slightly misleading, as not every cancer develops from an earlier stage, like glioblastoma (GBM IV), but it may be true for some colon tumors. To put the potential in relation to reducing overall cancer deaths, it may be worth mentioning that smoking cigarettes is the main single cause of cancer death and is therefore preventable even in low-income countries.
  4. Cerebrospinal fluid (CSF) could also be mentioned in the brain tumor section as being very useful in liquid biopsies of brain tumors (https://www.mdpi.com/2072-6694/13/21/5429).
Comments on the Quality of English Language

English is very good.

Author Response

Dear reviewer, many thanks for your valuable notes, improving the quality of our manuscript. We greatly appreciate your time, effort and expertise.

To address your suggestions:

Line 17: Perhaps avoid the word “delves”, since this appears to be used often by ChatGPT.

     Thank you for the suggestion, the beginning of the sentence was rewritten.  

Line 44: Consider rephrasing "Cancer is poised to become the primary cause of death worldwide." This claim is debatable. Even the reference cited only refers to a specific age group in high-income countries. So, worldwide, cancer is far from becoming the primary cause of death.

             Thanks for the important suggestion, the text was amended to avoid claiming boldly misleading fact, as CVD diseases will hold the prime slot for the future as well.

Line 46 "Detection of cancer before stage IV could potentially reduce cancer-related deaths by at least 15% within a five-year period" – This may be slightly misleading, as not every cancer develops from an earlier stage, like glioblastoma (GBM IV), but it may be true for some colon tumors. To put the potential in relation to reducing overall cancer deaths, it may be worth mentioning that smoking cigarettes is the main single cause of cancer death and is therefore preventable even in low-income countries.

           Thanks for the relevant suggestion, we have rewritten the sentence, to  mention that it is a model situation and applied for a certain cancers, encouraging readers to read an interesting cited study.   

Cerebrospinal fluid (CSF) could also be mentioned in the brain tumor section as being very useful in liquid biopsies of brain tumors (https://www.mdpi.com/2072-6694/13/21/5429).

          Thank you for the suggestion, this important information was added to the main text in tumor section as well as very informative review cited. 

Reviewer 2 Report

Comments and Suggestions for Authors

The manuscript focused Current Challenges Of Methylation-Based Liquid Biopsies In Cancer Diagnostics represents a technically correct and timely relevant manuscript on a hot topic for the clinical management of tumor patients. In my opinion, some moderate modifications should be approached to improve the readability of the manuscript on this journal.

- In the introduction section, please, could the authors overview the molecular mechanism behind epigenetic modifications in tumor cells?

- In the manuscript, please, could the authors clinically investigate the role of methylation based modifications in tumor patients? In my opinion, the authors should better consider the clinical window where methylation changes may be applied

- In the manuscript ,please, could the authors identify the most impacting challenges for the clinical implementation of methylation based approaches in clinical practice?

Comments on the Quality of English Language

Moderate english revision

Author Response

Dear reviewer, many thanks for your valuable notes, improving the quality of our manuscript. We greatly appreciate your time, effort and expertise.

To address your suggestions:

  •  In the introduction section, please, could the authors overview the molecular mechanism behind epigenetic modifications in tumor cells?

Thank you for this suggestion, we extended the mechanisms of abberant methylation in cancer cells, after careful consideration into the chapter 2 (DNA methylation and cancer)

  • In the manuscript, please, could the authors clinically investigate the role of methylation based modifications in tumor patients? In my opinion, the authors should better consider the clinical window where methylation changes may be applied

 Many thanks for your insightful note. The methylation involvement in timeframe of tumor involvement is very patient specific. In general, clinical window in cancer patients for methylation based liquid biopsy test is in later stages, where more epigenetic changes accumulate as well as more tumor cfDNA is released to body fluids. This problematic will be however not only adressed but also actively studied in basic research of our team, as this review is only first step in our research on liquid biopsies supported by international grant.         

  • In the manuscript ,please, could the authors identify the most impacting challenges for the clinical implementation of methylation based approaches in clinical practice?

 Thank you for valuable note, we extended the conclusion part of the review with mentioning main challenges of implementation of cancer specific liquid biopsy, tests.